# Non-Robust Feature Mapping in Deep Reinforcement Learning

**Ezgi Korkmaz** [1]

## Abstract

Adversarial perturbations to state observations can dramatically degrade the performance of deep reinforcement learning policies, and thus raise concerns regarding the robustness of deep reinforcement learning agents. A sizeable body of work has focused on addressing the robustness problem in deep reinforcement learning, and there are several recent proposals for adversarial training methods in the deep reinforcement learning domain. In our work we focus on the robustness of state-of-the-art adversarially trained deep reinforcement learning policies and vanilla trained deep reinforcement learning polices. We propose two novel algorithms to map non-robust features in deep reinforcement learning policies. We conduct several experiments in the Arcade Learning Environment (ALE), and with our proposed feature mapping algorithms we show that while the state-of-the-art adversarial training method eliminates a certain set of non-robust features, a new set of non-robust features more intrinsic to the adversarial training are created. Our results lay out concerns that arise when using existing state-of-the-art adversarial training methods, and we believe our proposed feature mapping algorithm can aid in the process of building more robust deep reinforcement learning policies.

## 1. Introduction

Learning reasonable representations from high dimensional raw data became possible with the introduction of deep neural networks. Recent successes of deep neural networks made several complex tasks achievable in many domains e.g. image recognition Sutskever et al. (2014), natural language processing (Hannun et al., 2014), self learning systems (Mnih et al., 2015). In particular, the utilization of deep neural networks as function approximators in reinforcement learning made it possible to learn policies for large state or action space Markov Decision Processes (MDP)s.

While the success of deep neural networks has grown several concerns have been raised related to robustness in the presence of specifically crafted adversarial perturbations (Goodfellow et al., 2015; Szegedy et al., 2014; Ilyas et al., 2019). To address the sensitivity problem several works have focused on making deep neural classifiers more robust to these specific perturbations by training with inputs modified via these perturbations (Madry et al., 2018; Goodfellow et al., 2015).

Several studies have demonstrated that the lack of robustness to adversarial perturbations observed in deep neural classifiers is also present in deep reinforcement learning policies (Huang et al., 2017; Kos & Song, 2017; Korkmaz, 2020; 2021b; 2022a). These concerns about the resilience of deep reinforcement learning were targeted by a line of work focused on building robust policies (Pinto et al., 2017; Pattanaik et al., 2018; Zhang et al., 2020). In our paper our aim is to try to answer the following questions: (i) Are there any vulnerability differences between the state-of-the-art adversarially trained deep reinforcement learning policies and vanilla trained deep neural policies?, and (ii) Do non-robust features still exist in different forms in state-of-the-art adversarially trained deep reinforcement learning policies? To fundamentally understand the sensitivities of deep reinforcement learning policies in our work we focus on adversarial training in deep reinforcement learning and make the following contributions:

- We propose two novel algorithms to visualize non-robust features in deep reinforcement learning policies.

- We conduct several experiments in the Arcade Learning Environment (ALE) with vanilla and adversarially trained deep reinforcement learning policies.

- We show that the state-of-the-art adversarial training method eliminates non-robust features intrinsic to the vanilla training techniques. However, we found that the state-of-the-art adversarial training method creates a new set of non-robust features intrinsic to adversarial training.

The complete version of this paper is published in Korkmaz (2021d). See the full version of the paper for more com-

---

[1] Ezgi Korkmaz <ezgikorkmazmail@gmail.com>

*Proceedings of the 37th International Conference on Machine Learning*, PMLR 108, 2021. Copyright 2021 by the author(s).

prehensive analysis on the vulnerabilities of state-of-the-art adversarial training. More diverse issues introduced by adversarial training in deep reinforcement learning are also discussed in Korkmaz (2021a;c; 2023).

## 2. Background and Preliminaries

### 2.1. Reinforcement Learning

A Markov Decision Process (MDP) is represented as a tuple $(\mathcal{S}, \mathcal{A}, r, \mathcal{P}, \gamma, \rho_0)$ where $\mathcal{S}$ represents a finite set of states, $\mathcal{A}$ represents a finite set of actions, $r$ represents the reward function, $\mathcal{P}$ represents the Markovian transition kernel, $\gamma$ represents the discount factor and $\rho_0$ represents the initial state distribution. The aim is to learn a value for each state-action pair to determine the expected discounted cumulative rewards to be obtained if action $a$ is executed in state $s$. This is achieved by learning the state-action value function $Q(s, a) = \mathbb{E}_{a \sim \pi(\cdot|s)}[\sum_{t=0}^{\infty} \gamma^t r(s_t, a_t) | s_0 = s, a_0 = a]$. Once the state-action value function is learnt the optimal policy $\pi^*(a|s)$ is given by taking the action $a^*(s) = \arg\max_a Q(s, a)$ in state $s$.

### 2.2. Adversarial Perturbations and Adversarial Training

Goodfellow et al. (2015) introduced a computationally inexpensive way to produce adversarial examples based on linearly approximating the loss function around a given input,

$$x_{\text{adv}} = x + \epsilon \cdot \frac{\nabla_x J(x, y)}{||\nabla_x J(x, y)||_p}, \quad (1)$$

where $x$ is a given input, $y$ is the output labels, and $J(x, y)$ is the cost function used to train the network. Note that perturbations computed in this way are bounded by $\epsilon$. Kurakin et al. (2016) focused on an iterative search using the fast gradient sign method introduced by Goodfellow et al. (2015).

$$x_{\text{adv}}^0 = x, \quad (2)$$
$$x_{\text{adv}}^{N+1} = \text{clip}_\epsilon(x_{\text{adv}}^N + \alpha \text{sign}(\nabla_x J(x_{\text{adv}}^N, y))) \quad (3)$$

Madry et al. (2018) referred to this class of iterative methods as projected gradient descent (PGD), and gave a theoretical justification for training with inputs modified by such perturbations using robust optimization theory.

### 2.3. Robustness in Deep Reinforcement Learning

Investigation into the robustness of deep reinforcement learning policies was first conducted by Huang et al. (2017) and Kos & Song (2017). In these studies the authors show the lack of robustness of deep reinforcement learning policies to fast gradient sign method produced perturbations. Pattanaik et al. (2018) propose to increase the probability

of worst possible action (i.e. $\arg\min_a Q(s, a)$) in a given state to achieve higher degradation of the agent's performance. Pinto et al. (2017) models the interaction between the adversary and the agent as a zero-sum Markov game and proposes a training strategy to compute a robust policy for the agent. More recently, Zhang et al. (2020) propose to model this problem as a State-Adversarial Markov Decision Process (SA-MDP). In this study, the authors claim that their proposed SA-MDP model and algorithm obtain theoretically justified robust policies towards both natural errors and adversarial perturbations. Recently, Korkmaz (2022a) showed that shared adversarial features learnt by deep reinforcement learning policies exist across MDPs. Quite recently, Korkmaz (2023) questioned the distinctions and similarities between natural directions that are intrinsic to the MDP and adversarial directions in terms of degradation they cause on the policy performance and their perceptual similarities to the base state observations. The results reported in Korkmaz (2023) while questioning the robustness definition in certified state-of-the-art adversarial training techniques, further demonstrate that the certified adversarial training techniques hurt generalization capabilities of deep reinforcement learning policies[1]. More explanations and a comprehensive review on generalization in deep reinforcement learning can be found in this recent survey (Korkmaz, 2024).

## 3. Mapping Non-Robust Features in Deep Reinforcement Learning Policies

In this paper we aim to seek answers for the following questions:

- What are the susceptibility differences between state-of-the-art adversarially trained deep reinforcement learning policies and vanilla trained deep reinforcement learning policies?

- Do the sensitivities of deep reinforcement learning policies shifts from worst-case $\ell_p$-norm bounded perturbations towards different directions in the input with adversarial training?

- Does adversarial training create a new set of non-robust features while eliminating the existing ones?

In this section we propose two different methods to visualize vulnerabilities of deep reinforcement learning policies to their input observations. First, we describe our proposed method of feature vulnerability mapping KMAP in detail.

---

[1]There are also some recent studies focusing on the robustness problems in deep neural policies that can learn without a reward function provided by the MDP (i.e. inverse reinforcement learning and imitation learning). See these studies for more details (Korkmaz, 2022b;c).

---

**Algorithm 1** HMAP Feature vulnerability mapping

**Input:** State-action value function $Q(s,a)$, actions $a$, states $s$, policy $\pi(s,a)$, $T_d$ size of the dimension $d$ of the state $s$, and $s(i,j)$ is the value of the $i,j$-th pixel in state $s$.
**Output:** Visual weakness mapping function $\mathcal{H}(i,j)$
$s_{\text{aug}} = s$
**for** $i = 1$ **to** $T_1$ **do**
  **for** $j = 1$ **to** $T_2$ **do**
    $s_{\text{aug}}(i,j) = 0$
    $\pi(s,a) = \text{softmax}(Q(s,a))$
    $\pi(s_{\text{aug}},a) = \text{softmax}(Q(s_{\text{aug}},a))$
    $\mathcal{H}(i,j) \mathrel{+}= -\sum_{a \in A} \pi(s,a) \log(\pi(s_{\text{aug}},a))$
    $s_{\text{aug}} = s$
  **end for**
**end for**
**Return:** $\mathcal{H}(i,j)$

---

**Algorithm 2** KMAP Feature vulnerability mapping

**Input:** State-action value function $Q(s,a)$, actions $a$, states $s$, $T_d$ size of the dimension $d$ of the state $s$, and $s(i,j)$ is the value of the $i,j$-th pixel in state $s$.
**Output:** Visual weakness mapping function $\mathcal{K}(i,j)$
$s_{\text{aug}} = s$
**for** $i = 1$ **to** $T_1$ **do**
  **for** $j = 1$ **to** $T_2$ **do**
    $s_{\text{aug}}(i,j) = 0$
    $a^*_{\text{aug}} = \text{argmax}_a\ Q(s_{\text{aug}},a)$
    $a^* = \text{argmax}_a\ Q(s,a)$
    $\mathcal{K}(i,j) \mathrel{+}= Q(s,a) - Q(s,a^*_{\text{aug}})$
    $s_{\text{aug}} = s$
  **end for**
**end for**
**Return:** $\mathcal{K}(i,j)$

---

To be able to visualize weaknesses we record the drop in the state-action value $Q(s,a)$ caused by setting each pixel in $s$ to zero one at a time. In particular, let $Z_{i,j} : \mathcal{S} \to \mathcal{S}$ be the function which sets the $i,j$ coordinate of $s$ to zero and leaves the other coordinates unchanged. We define,

$$\mathcal{K}(i,j) = Q(s,a^*) - Q(s, \arg\max_a Q(Z_{i,j}(s),a)). \quad (4)$$

Note that the difference in Equation 4 represents the drop in the $Q$-value in state $s$, when taking the optimal action for the state $Z_{i,j}(s)$. Therefore, $\mathcal{K}(i,j)$ aims to measure the drop in the $Q$-values of the reinforcement learning policy with respect to individual pixel changes. In other words, $\mathcal{K}(i,j)$ is a mapping of features to an importance metric determined by the deep reinforcement learning policy. We describe our proposed KMAP method in detail in Algorithm 2.

As a natural point of comparison we propose another algorithm HMAP to visualize input based vulnerabilities. In particular, HMAP is based on measuring the effect of each individual pixel on the decision of the deep reinforcement learning policy by measuring the cross-entropy loss between $\pi(s,a)$ and $\pi(Z_{i,j}(s),a)$.

$$\mathcal{H}(i,j) = -\sum_{a \in A} \pi(s,a) \log(\pi(Z_{i,j}(s),a)) \quad (5)$$

where we compute the policy $\pi(s,a)$ via the softmax of the state-action value function $Q(s,a)$,

$$\pi(s,a) = \frac{e^{Q(s,a)/T}}{\sum_{a \in A} e^{Q(s,a)/T}}. \quad (6)$$

Note that $T$ represents the temperature constant. We describe the HMAP method in detail in Algorithm 1.

## 4. Results on KMAP and HMAP

The vanilla trained deep reinforcement learning policies are trained via Double Deep Q-Network (DDQN) Wang et al. (2016), and the the-state-of-the-art adversarially trained deep reinforcement learning policy is trained via State Adversarial Double Deep Q-Network (SA-DDQN) Zhang et al. (2020). Note that Zhang et al. (2020) uses a regularization term,

$$\mathcal{R}(\theta) = \sum_s \max\{ \max_{s \in \mathcal{D}(s)} \max_{a \neq a^*} Q_\theta(s_{\text{adv}}, a) \quad (7)$$
$$- Q_\theta(s_{\text{adv}}, a^*(s)), c\} \quad (8)$$

added in the temporal difference loss in $Q$-learning in training time where $\mathcal{D}(s) := \{s_{\text{adv}} | \, \|s_{\text{adv}} - s\|_\infty \leq \epsilon\}$. The deep reinforcement learning policies are trained in Arcade Learning Environment (ALE) proposed by Bellemare et al. (2013) in OpenAI version Brockman et al. (2016).

Figure 2 and Figure 1 show heatmaps of feature vulnerability mapping KMAP $\mathcal{K}(i,j)$ and HMAP $\mathcal{H}(i,j)$ for the state-of-the-art adversarially trained deep reinforcement learning policy and vanilla trained deep reinforcement learning policy for Freeway. We observe that while the KMAP $\mathcal{K}(i,j)$ pattern for the vanilla trained agent lies on the portion of the input where the optimal policy is executed by the agent, the KMAP $\mathcal{K}(i,j)$ for the adversarially trained deep reinforcement learning policy has a straightforward grid pattern. Based on these results, we hypothesize that adversarial training decouples vulnerability from the features relevant to the optimal policy learned by the agent. The decoupling of relevant features and vulnerability can be seen as a way in which adversarial training shifts the vulnerabilities of deep reinforcement learning policies.

While visual observation indicates very different vulnerability patterns for these two disjoint training strategies, we

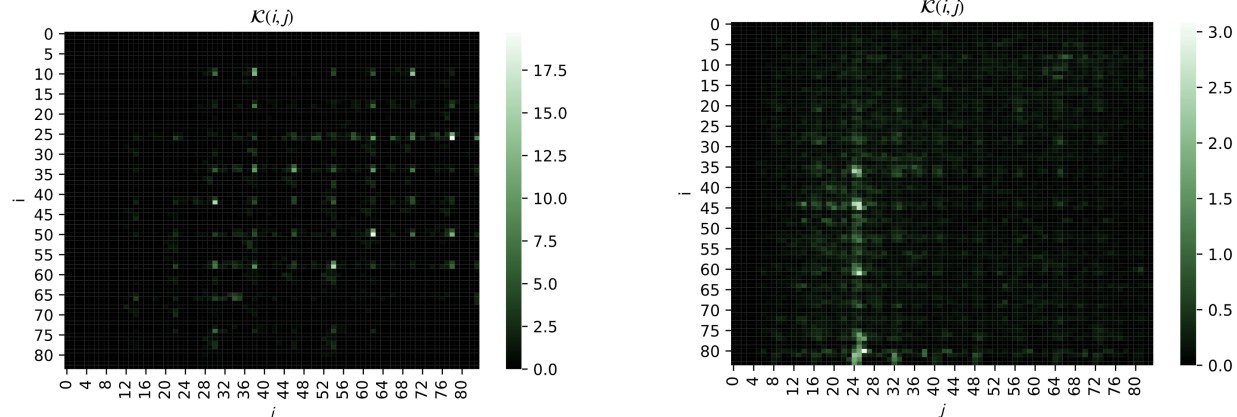

*Figure 1.* KMAP $\mathcal{K}(i,j)$ and HMAP $\mathcal{H}(i,j)$ heatmaps for state-of-the-art adversarially trained deep reinforcement learning policy and vanilla trained deep reinforcement learning policy in Freeway. Left: Adversarially trained (SA-DDQN). Right: Vanilla trained (DDQN).

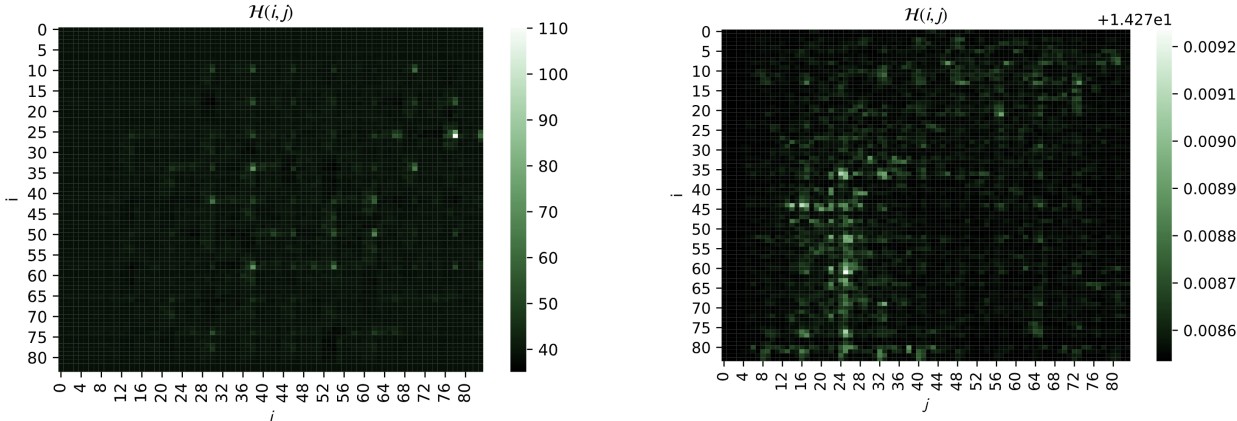

*Figure 2.* HMAP $\mathcal{H}(i,j)$ heatmaps for state-of-the-art adversarially trained deep reinforcement learning policy and vanilla trained deep reinforcement learning policy in Freeway. Left: Adversarially trained (SA-DDQN). Right: Vanilla trained (DDQN).

*Table 1.* Sparsity results of KMAP $\mathcal{K}(i,j)$ and HMAP $\mathcal{H}(i,j)$ for adversarially and vanilla trained deep reinforcement learning policies.

| Training Method | Vanilla Trained | Adversarially Trained | Vanilla Trained | Adversarially Trained |
| Sparsity | $S(\mathcal{K})$ | $S(\mathcal{K})$ | $S(\mathcal{H})$ | $S(\mathcal{H})$ |
| --- | --- | --- | --- | --- |
| Freeway | 53.7272 | 20.4641 | 83.9999 | 83.91587 |

also introduce a quantitative metric to compare the results of KMAP and HMAP for vanilla and adversarially trained agents. In particular, we use the ratio of the $\ell_1$ and $\ell_2$ norms to measure the sparsity via,

$$S(\mathcal{K}) = \frac{\|\mathcal{K}\|_1}{\|\mathcal{K}\|_2}. \qquad (9)$$

Here smaller values of $S(\mathcal{K})$ correspond to sparser vulnerability patterns. In Table 1 we show the sparsity results

respectively for KMAP $\mathcal{K}(i,j)$ and HMAP $\mathcal{H}(i,j)$ for adversarially trained deep reinforcement learning policies and vanilla trained deep reinforcement learning policies. We observe that for KMAP the vulnerability of adversarially trained models with respect to features are more sparse than the vanilla trained agents. The results for HMAP are more mixed, and it is often barely possible to detect the sparsity difference via $S(\mathcal{H})$. In general, KMAP $\mathcal{K}(i,j)$ provides a better estimation of sensitivity of deep reinforcement learning policies to individual pixel changes than HMAP $\mathcal{H}(i,j)$.

While KMAP $\mathcal{K}(i, j)$ captures the actual impact of the feature change on the decision of the deep reinforcement learning policy HMAP $\mathcal{H}(i, j)$ captures the difference between the softmax policy distributions $\pi(s, a)$ and $\pi(Z_{i,j}(s), a)$, which do not necessarily correspond to the decisions made by the neural policy.

## 5. Conclusion

In this paper we focused on investigating the vulnerabilities of deep reinforcement learning policies with respect to their inputs. We propose two different algorithms that we call KMAP and HMAP to detect vulnerabilities with respect to input in deep reinforcement learning policies. We compare the state-of-the-art adversarially trained neural policies and vanilla trained neural policies with our proposed methods KMAP and HMAP via several experiments in various environments. With the help of our proposed feature vulnerability mapping algorithm we found that while adversarial training removes sensitivity to certain features, it builds sensitivity towards a new set of features. We believe this work lays out the vulnerabilities of adversarially trained neural policies in a systematic way, and can be an initial step towards building robust and reliable deep reinforcement learning agents.

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
