# OpenReview forum: "Non-Robust Feature Mapping in Deep Reinforcement Learning"
_ICML.cc/2021/Workshop/AML — ICML 2021 Workshop AML Poster_

### Official Review · Reviewer_QG1i · 2021-06-20
**The results are interesting but unconvincing**

**Rating:** Accept
**Confidence:** 5

**Review:**

This paper proposed two feature mappings and visualized these features in DRL policies.  The paper further investigated the sparsity of these two mappings, and emphasize different vulnerabilities for these mappings. This paper provides some insightful ideas.

My main concern of this work is it over claims the results. This paper didn't provide the way on choosing the observation which weakens the credibility of this work.

Less curial problems:
1) Alg. 2 return K instead of K(i,j)
2) There are some small mistakes in Fig. 2

---

### Decision · Program_Chairs · 2021-06-21

**Decision:**

Accept (Poster)

**Comment:**

This paper proposed two feature mappings and visualized these features in DRL policies. The concerns of the reviewer could be further addressed.